# AP-Adapter: Improving Generalization of Automatic Prompts on Unseen Text-to-Image Diffusion Models

**Yuchen Fu**    **Zhiwei Jiang**[*]    **Yuliang Liu**    **Cong Wang**
**Zexuan Deng**    **Zhaoling Chen**    **Qing Gu**

State Key Laboratory for Novel Software Technology, Nanjing University, China

`yuchenfu@smail.nju.edu.cn, jzw@nju.edu.cn`
`{yuliangliu,cw,dengzx,zhaolingchen}@smail.nju.edu.cn`
`guq@nju.edu.cn`

## Abstract

Recent advancements in Automatic Prompt Optimization (APO) for text-to-image generation have streamlined user input while ensuring high-quality image output. However, most APO methods are trained assuming a fixed text-to-image model, which is impractical given the emergence of new models. To address this, we propose a novel task, model-generalized automatic prompt optimization (MGAPO), which trains APO methods on a set of known models to enable generalization to unseen models during testing. MGAPO presents significant challenges. First, we experimentally confirm the suboptimal performance of existing APO methods on unseen models. We then introduce a two-stage prompt optimization method, AP-Adapter. In the first stage, a large language model is used to rewrite the prompts. In the second stage, we propose a novel method to construct an enhanced representation space by leveraging inter-model differences. This space captures the characteristics of multiple domain models, storing them as domain prototypes. These prototypes serve as anchors to adjust prompt representations, enabling generalization to unseen models. The optimized prompt representations are subsequently used to generate conditional representations for controllable image generation. We curate a multi-modal, multi-model dataset that includes multiple diffusion models and their corresponding text-image data, and conduct experiments under a model generalization setting. The experimental results demonstrate the AP-Adapter's ability to enable the automatic prompts to generalize well to previously unseen diffusion models, generating high-quality images.

## 1 Introduction

Automatic Prompt Optimization (APO) in text-to-image generation aims to simplify the user input process while ensuring that the generated images exhibit both semantic consistency and aesthetic quality [9, 22, 24]. APO addresses the limitation of current text-to-image models, which often heavily rely on manually crafted prompts for generating high-quality images. The direct use of natural language descriptions as inputs often results in suboptimal outcomes.

APO methods offer effective ways to directly utilize natural language descriptions for text-to-image diffusion models, which can be categorized into two paradigms: Text Optimization and Representation Optimization. Text Optimization entails generating refined prompts directly by fine-tuning the language model [8, 26]. On the other hand, Representation Optimization involves refining prompts within the feature space of the text encoder of the diffusion model [37].

---

[*]Corresponding author

38th Conference on Neural Information Processing Systems (NeurIPS 2024).

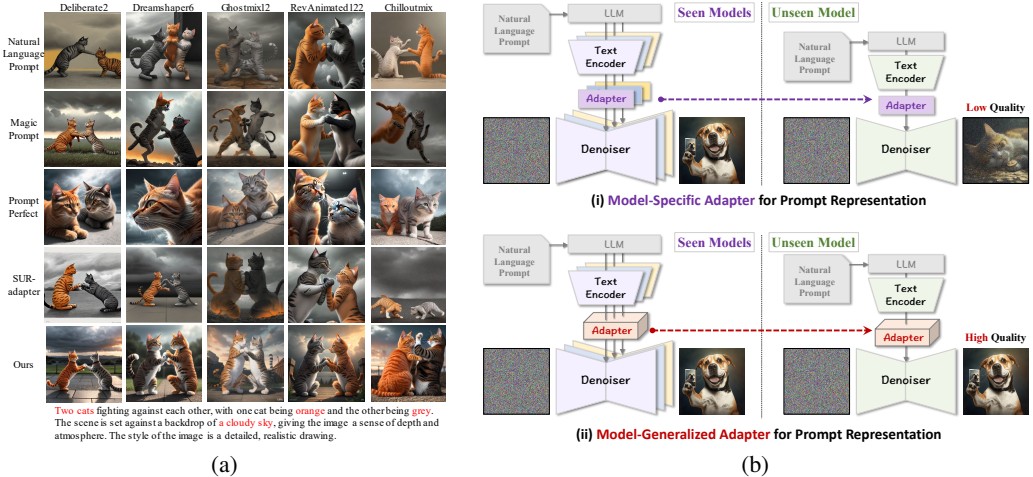

(a)             (b)

Figure 1: (a) Comparison of text-to-image generation among different APO methods on various unseen fine-tuned diffusion models. (b) Comparison between model-specific and model-generalized adapters under the model-generalized setting.

As the advancements like LoRA [13] and DreamBooth [27] have simplified the training of diffusion models with personalized data, an increased number of diffusion models have emerged. The emergence of new models constantly requires us to conduct APO for new models and fresh prompt data. This process is overly labor-intensive and time-consuming. However, directly applying APO method specific to seen models to unseen models may result in poor generalization. As shown in the Figure 1a, Text optimization methods like MagicPrompt [7] and PromptPerfect [16] enhance image quality but sacrifice semantic consistency. Representation optimization methods, exemplified by SUR-adapter [37] maintain semantic consistency but show limited improvement in image quality.

In this work, we address the model generalization scenario of Automatic Prompt Optimization (APO) and introduce the novel task of model-generalized automatic prompt optimization (MGAPO). The objective of MGAPO is to train an APO model on a limited set of known diffusion models and enable automatic prompt generalization for unseen diffusion models. MGAPO presents several challenges. First, implementing MGAPO requires incorporating extensive model information, which is absent in existing synthetic image datasets [37, 31]. Second, MGAPO must ensure both the semantic consistency and aesthetic quality of the generated images. Third, to achieve good generalization on unseen diffusion models, the model-related knowledge must be appropriately addressed during training, either retained or discarded.

To train an APO model with generalization capabilities, we frame the task of MGAPO as a variant of domain generalization. We initially collected and annotated a multi-modal multi-domain dataset for training and testing. The dataset comprises natural language prompts, manually crafted prompts, high-quality images, and multiple stable diffusion (SD) checkpoints, serving as domain information. We then propose the Automatic Prompt Adapter (AP-Adapter), which is a two-stage framework. In the first stage, we leverage the ICL capability of large language models (LLMs) [23, 30] to generate automatic keyword prompts similar to manually crafted prompts [1, 34]. These prompts are then fed into the diffusion model's text encoder. In the second stage, we achieve model generalization using an adapter-based approach. The distinction between model-specific [37] and model-generalized adapters is illustrated in Figure 1b. We decode domain information from images [4, 35] and distill this information into domain prototypes. These prototypes serve as anchors to adjust the prompt representation, providing rich domain information for prompt adaptation. By further aligning with the representation specific to manual prompts, the adapted representation can simultaneously possess the ability for both semantically consistent and aesthetically pleasing generation.

Our main contributions are as follows:

- We explore model-generalized automatic prompt optimization (MGAPO), targeting the effectiveness of automatic prompts on unseen models, addressing a challenging aspect of domain generalization.

- We propose AP-Adapter, the first method to address MGAPO, leveraging in-context learning of large language model for text optimization in the first stage and constructing an enhanced representation space by leveraging inter-model differences in the second stage.
- We collect and annotate a multi-modal, multi-domain dataset for training and evaluation, including high-quality images, manually crafted prompts, natural language prompts, and model information.
- Our extensive experiments on this dataset demonstrate Adapter's ability to enable the automatic prompts to generalize well to previously unseen diffusion models.

## 2   Related Work

**Automatic Prompt Optimization for Text-to-Image Models.**Prompt Optimization is essential for accurate semantic comprehension and high-quality image generation. Hao et al. [8] proposed a two-stage training approach. Initially, they supervised fine-tuned a language model using manual prompts. In the second stage, reinforcement learning, such as Proximal Policy Optimization (PPO), was employed to maximize the aesthetic and relevance scores of generated images. Rosenman [26] introduced the Neurologic Decoding method to generate more diverse and personalized prompts. Zhong et al. [37] improved image quality by aligning natural language descriptions with manually designed prompts in the feature space of CLIP's text encoder through the optimization of a low-parameter adapter. However, challenges in adapting pre-trained adapters to out-of-domain data and overlooking diversity across Stable Diffusion (SD) models persist. Noteworthy products, including Promptperfect and MagicPrompt, offer similar functionality, but our usage reveals limitations in their effective generalization across diverse SD models.

**Domain Generalization.**Domain generalization aims to distill domain-invariant knowledge from a source domain, facilitating seamless model generalization to unknown domains. Advanced strategies leverage diversity within known domains to construct an enhanced feature space, aiding representation of unknown data using knowledge gained from known domains. DGSS [15] preserves style features from known domains as bases for generating new stylized images, promoting generalization to unknown domains. StyleAdv [5] introduces adversarial perturbations to image styles, enhancing model robustness to unknown image styles through adversarial training. Other works, such as [29, 3, 19], fully exploit the adapter module for domain generalization on pre-trained models.

## 3   Method

### 3.1   Task Definition

Similar to domain generalization, MGAPO aims to achieve generalization to unseen models in the target domain by training the APO model on K source domain models. Formally, we introduce notations and formalize the task as follows. Given a diffusion model (i.e., Stable Diffusion 1.5) $f(x; \Theta)$ with $K$ source checkpoints $\mathcal{S} = \{\Theta_1, \Theta_2, ..., \Theta_K\}$, the $k$-th source checkpoint contains $N_k$ image-text pairs $(n_i^k, m_i^k, I_i^k)_{i=1}^{N_k}$, where $n_i^k$ represents natural language descriptions, $m_i^k$ denotes manually designed prompts, and $I_i^k$ is the corresponding image. Each checkpoint has the same architecture as $f(x; \Theta)$ but with different parameters. Therefore each source checkpoint with its text-image pairs can be viewed as a source domain. In the testing phase, unseen target checkpoints $\mathcal{T} = \{\Theta_{K+1}, \Theta_{K+2}, ..., \Theta_{K+T}\}$ with natural language descriptions $(n_i^t)_{i=1}^{N_t}$ are provided. Each target checkpoint can be viewed as a target domain. Our objective is to leverage source-domain checkpoints to create a pipeline capable of generalizing to any unseen target-domain checkpoints, generating coherent and aesthetically pleasing images for any given natural language descriptions.

### 3.2   Overview

As illustrated in Figure 2, our method comprises two stages of optimization for text prompts, corresponding to text-based optimization and representation-based optimization methods, respectively.

In the first stage, we leverage the zero-shot reasoning capability of large language models $f_{llm}$ by employing in-context learning [21] to guide the generation of reliable prompts. Examples for in-context learning are obtained through similarity retrieval. In this step, natural language prompts

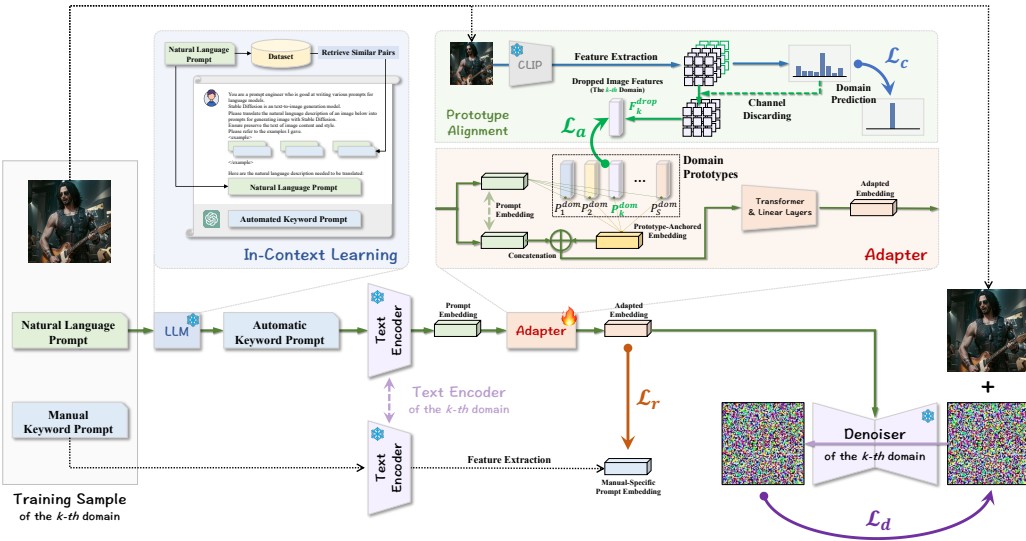

Figure 2: Illustration of our proposed AP-Adapter.

are automatically translated into keyword prompts, which are then fed into the text encoder of the SD model $f_{clip}^T$ to obtain text representations.

In the second stage, we improve automatic keyword prompts by decoding domain-specific content from images and aligning it with manual prompts. To handle image unavailability during inference, we maintain prototypes in training to preserve domain-specific details. The adapter module then integrates information from text and prototypes, aligning its output representation with manual prompts in the feature space.

Specifically, we freeze the parameters of the text encoder $f_{clip}^T$ and denoiser $f_{den}$ in diffusion model. During training, for a specific domain's image-text pair $(n_i^k, m_i^k, I_i^k)$, we load the corresponding diffusion model parameters $f(x; \Theta_k)$ to extract features and denoise. The CLIP image encoder $f_{clip}^I$ is introduced to extract image features, with the image encoder parameters also frozen.

### 3.3 ICL-Based Prompt Rewriting

Previous work [21] has proved the demonstrations that are semantically similar to the test example exhibit higher performance in context learning. We employ a pre-trained sentence transformer [25] to calculate the cosine similarity between input and natural prompts in dataset. For each input natural prompts, we select the top $k$ most similar demonstrations from dataset. These pairs of natural language prompts and manually crafted prompts, along with the input natural prompts, are embedded into the prompt template (depicted on the left side of Figure 2). The prompt template is then feed into GPT3.5 to obtain preliminary optimized keyword prompt. Compared to natural prompts, keyword prompts are closer in the semantic space of CLIP to manually crafted prompts. However, they are still insufficient for achieving multi-model generalization. Therefore, we introduce a prototype-based adaptive approach for automatic keyword prompts.

### 3.4 Prototype-Based Prompt Adaptation

In this section, we introduce how the adapter module leverages prototype learning to obtain model generalization capabilities.

**Alignment of Domain Prototypes.** The parameter difference of models is the main factor leading to the domain shift, which limits the generalization ability of the optimized prompts. Parameter differences are encoded into the image during the process of generating images from text, resulting in the images exhibiting different domain variations. Therefore, we propose domain prototypes, which align with the domain-specific information decoded from the image.

In detail, we employ intermediate features $F_l^k \in \mathbb{R}^{C \times H \times W}$ with the $l$-th CLIP image encoder layer $f_{clip}^I$ from image $I_i^k$. Here, $C$ is the number of feature channels, while $H$ and $W$ denote the feature map's dimensions. Domain prototypes, denoted as $\{P_k^{dom} \in \mathbb{R}^{N_p \times D}\}_{k=1}^K$, aim to align with the domain-specific information from image features to enhance automatic keyword prompt representations. $N_p$ denotes the number of prototypes, and $D$ is the text token dimension.

Identifying domain-specific information in image features is challenging due to the higher dimensionality of image features $C$ compared to text features $D$. Inspired by DomainDrop [6], we selectively drop non-informative channels in the feature space while retaining domain-specific information.

To drop domain-insensitive feature channels, we use a domain discriminator $f_d(\cdot)$ that predicts the image's domain based on intermediate layer features. This discriminator, comprising a global average pooling (GAP) layer and a fully-connected (FC) layer, outputs the domain label $\hat{y}$ and computes cross-entropy loss for optimization:

$$\mathcal{L}_c = -\frac{1}{K} \sum_{k=1}^K \hat{y} \log(f_d(F_l^k)) \tag{1}$$

where $K$ is the number of domains. Weights in the FC layer quantify each channel's contribution to domain discrimination. For the c-th channel in the feature map, its contribution is calculated as:

$$score_c = W_c^{\hat{y}} \cdot \text{GAP}(F_l^k) \tag{2}$$

where $W^{\hat{y}} \in \mathbb{R}^C$ is the weight in the fully connected layer when predicting the domain label $\hat{y}$ correctly. A higher score indicates a greater contribution of the channel to domain discrimination. Subsequently, we can generate a binary mask $M$ through the weighted random selection algorithm [12].

$$M_c = \begin{cases} 1, & \text{if } c \in \text{Top}(\{v_1, v_2, ..., v_C\}, D) \\ 0, & \text{otherwise} \end{cases} \tag{3}$$

where $v_c = r_c^{1/score_c}$ is a computed by a random number $r_c \in (0, 1)$, $\text{Top}(D)$ denotes the represents the top $D$ items sorted by the values of $v_c$, $C$ and $D$ are the dimensions of image features and text features, respectively.

By dropping the domain-insensitive channels of the image feature channels, we enhance the representation of domain information in the features and ensure consistency in the number of feature channels between images and text. To prevent drastic changes in image domain during the training process, we employ momentum updates for smoothing:

$$F_k^{drop'} = \alpha F_k^{drop} + (1 - \alpha)F_k^{drop'} \tag{4}$$

where $\alpha$ is a hyperparameter used to adjust the smoothness, $F_k^{drop}$ and $F_k^{drop'}$ are the image features after channel dropout and the features at the next step, respectively.

We initialize the domain prototypes with standard normal distribution and then align the domain prototypes with the dropped image features:

$$\mathcal{L}_a = \text{KL}(P_k^{dom}, F_k^{drop}) \tag{5}$$

where $\text{KL}(\cdot, \cdot)$ means KL divergence.

**Adaptation of Prompt Representation.** We leverage all learned domain prototypes to achieve generalization to unseen models. We use the wasserstein distance $\text{dist}(\cdot, \cdot)$ to measure the distance between prompt embeddings $f_{clip}^T(f_{llm}(n_i^k)) \in \mathbb{R}^{L \times D}$, where $L$ is the prompt token length, and domain prototypes $P_s^{dom}$ across all source domains:

$$d_k = \text{dist}(P_k^{dom}, f_{clip}^T(f_{llm}(n_i^k))) \tag{6}$$

where $d_k$ represents the distance between the $k$-th domain prompt and the text token embedding. Then, we calculate the reciprocal of $d_k$ to determine the weight corresponding to each domain prompt:

$$\eta_k = \text{softmax}(\frac{1}{1 + d_k}) \tag{7}$$

where the softmax operation ensures that the sum of all weights is 1. We performed a weighted sum of all domain prototypes across domains to obtain the final Prototype-Anchored Embedding:

$$P_a = \sum_{k=1}^{K} \eta_k P_k^{dom} \tag{8}$$

Token embeddings of keyword prompts and mixed domain prototypes are concatenated for further training. We use aligned domain prototypes as anchors to adjust the representation of prompts, obtaining generalization capabilities on unseen domains.

**Alignment of Adapted Representation.** We introduce a simple yet effective adapter $g_{ada}$ to aggregate and consolidate relevant information from all tokens, generating a conditional encoding representation for text-to-image generation. Comprising two transformer encoder layers and one linear layer, the adapter's linear layer parameters are initialized to 0 for stable training, following a proven approach [36].

The adapter's output is denoted as:

$$q_i^k = g_{ada}(\text{concat}[f_{clip}^T(f_{llm}(n_i^k)), P_a]) \tag{9}$$

Our goal is to make the representation of $q_i^s$ as similar as possible to the representation of manually designed prompts. Therefore, we use mean squared error loss to align the representation of natural prompts and manual prompts.

$$\mathcal{L}_r = \frac{1}{N} \sum_{i=1}^{N} (f_{clip}^T(m_i^k) - q_i^k)^2 \tag{10}$$

### 3.5 Model Training

**Image Denoising.** During training phase, we need to ensure that the learned parameters do not compromise the denoising model's generation. We adopt the denoising loss from the diffusion model [28, 11] to guarantee this aspect.

$$\mathcal{L}_d = \mathbb{E}\|\epsilon - \epsilon_\theta(\alpha_t x_0 + \beta_t \epsilon, t, q_i^k)\|_2^2 \tag{11}$$

where $\epsilon_\theta$ represents the denoising model, $\epsilon$ represents the noise added to the input image $x_0$, $t$ represents the time step, and $\alpha_t x_0 + \beta_t \epsilon$ is the result of adding noise to the image at time step $t$.

**Alternating Training.** To optimize training efficiency, we employ two strategies. Firstly, considering the variation in diffusion model parameters during training, we streamline the process by sequentially training on all data from each domain within a single epoch. This minimizes the time spent on model switching. Secondly, to prevent interference between the learning of style prompts and adapters, we adopt an alternating training approach. Specifically, after completing an iteration across all data from each domain, we freeze either the adapter or domain prototypes parameters for the subsequent training epoch, repeating this cycle. Therefore, the loss used for training the domain prototypes is as follows:

$$\mathcal{L}_{pro} = \gamma_1 \mathcal{L}_a + \gamma_2 \mathcal{L}_c \tag{12}$$

while the loss used for training the adapter is denoted as:

$$\mathcal{L}_{ada} = \gamma_3 \mathcal{L}_r + \gamma_4 \mathcal{L}_d \tag{13}$$

where $\gamma_1, \gamma_2, \gamma_3, \gamma_4$ are the loss coefficients, ranging from [0, 1].

Our training pipeline is outlined in Appendix A. During the inference phase, we leverage the acquired domain prototypes and adapter for accurate predictions.

## 4 Experiments

### 4.1 Dataset
In this section, we introduce the process of collecting and creating the multi-modal multi-domain dataset for MGAPO.

Table 1: Evaluation of the generated images on the target domain data using diverse prompt optimization methods. AKP denotes the automatic keyword prompt generated by ICL-based prompt rewriting.

| Methods | Semantic Consistency | | | | Image Quality | | |
|---|---|---|---|---|---|---|---|
| | Color | Shape | Texture | Blipscore | Aesthetic Score | ImageReward | HPS |
| MagicPrompt | 0.438 | 0.395 | 0.432 | 0.297 | 6.154 | 0.066 | 0.207 |
| PromptPerfect | 0.433 | 0.401 | 0.425 | 0.302 | 6.249 | 0.124 | 0.211 |
| Promptist | 0.439 | 0.398 | 0.427 | 0.292 | 6.000 | 0.089 | 0.202 |
| SUR-adapter | 0.472 | 0.413 | 0.449 | 0.325 | 6.009 | 0.286 | 0.198 |
| AKP | 0.456 | 0.401 | 0.437 | 0.305 | 6.113 | 0.253 | 0.213 |
| AKP + SUR-adapter | 0.442 | 0.407 | 0.441 | 0.315 | 6.158 | 0.233 | 0.210 |
| Ours | **0.477** | **0.422** | **0.452** | **0.332** | **6.384** | **0.427** | **0.218** |
| Manual Prompts (GT) | / | / | / | 0.400 | 6.564 | 0.782 | 0.223 |

**Data Collection.** We sourced high-quality images and personalized SD checkponts from the CIVITAI community[2]. We collected 47,695 image-text pairs gathered from various checkpoints, ensuring privacy protection. Further analysis of our dataset is provided in the Appendix B.1.

**Data Cleaning.** To ensure quality and semantic consistency, we employed Clipscore [10] and the aesthetic predictor[3]. Qualifying samples, totaling 25,395 image-text pairs, were selected based on aesthetic and clip scores surpassing their respective 25th percentiles.

We enriched the dataset using LLaVA 1.5 [20], a multimodal language model, to generate detailed natural language descriptions. We utilized instructions from [2], guiding the generation with, "Describe this image and its style in a very detailed manner." Then, we selected the top 3 generated sentences with the highest relevance to the image content as natural language prompts.

Our dataset contains a total of over 5000 checkpoints obtained through fine-tuning based on SD 1.5. However, most checkpoints have fewer than 10 corresponding image-text pairs. To ensure fairness in learning, we sorted the checkpoints based on the number of corresponding image-text pairs and selected the top 40 checkpoints as the source domain data and checkpoints ranked 41 to 100 as the target domain data. The source domain encompasses 7075 samples, whereas the target domain comprises 3064 samples. The remaining data is used as demonstrations for similarity retrieval in ICL-based prompt rewriting.

## 4.2 Implementation Details

During the training phase, we retrieve 5 pairs of natural language prompts and manually designed prompts as demonstrations for ICL from the dataset. We instruct GPT3.5 to generate 5 keyword prompts based on the input natural language example. By combining these 5 keyword prompts with their corresponding manual prompts and images, we augment the data used to train the model-generalized adapter.

For the model's parameter settings, since the source domain data contains 40 checkpoints, the number of domain prototypes $S$ is set to 40. The coefficients $\gamma_1, \gamma_2, \gamma_3, \gamma_4$ for the loss functions are 0.01, 1.0, 0.001 and 1.0, respectively. Figure 9 shows the accuracy of the domain discrimination using the image features output from various layers of the CLIP image encoder. We use the image features from the 18-th layer, where the accuracy is the highest, as the input for the domain discriminator.

## 4.3 Evaluation Metric

The evaluation of our method on the target domain data encompasses both semantic consistency between text and images and image quality. For semantic consistency, we used Blipscore [17, 18] and followed the part on attribute binding in T2I-CompBench [14]. For image quality, we used aesthetic predictor, ImageReward [33], and Human Preference Score(HPS) [32]. Furthermore, we recruited human evaluators to assess the aesthetics of the images and the consistency between text and images.

For the metrics of color, shape, and texture, we use the model parameters in the target domain from our dataset and the text provided by T2I-CompBench as input. For other metrics, we use the model parameters and natural prompts in the target domain from our dataset.

---

[2]https://civitai.com/
[3]https://github.com/christophschuhmann/improved-aesthetic-predictor

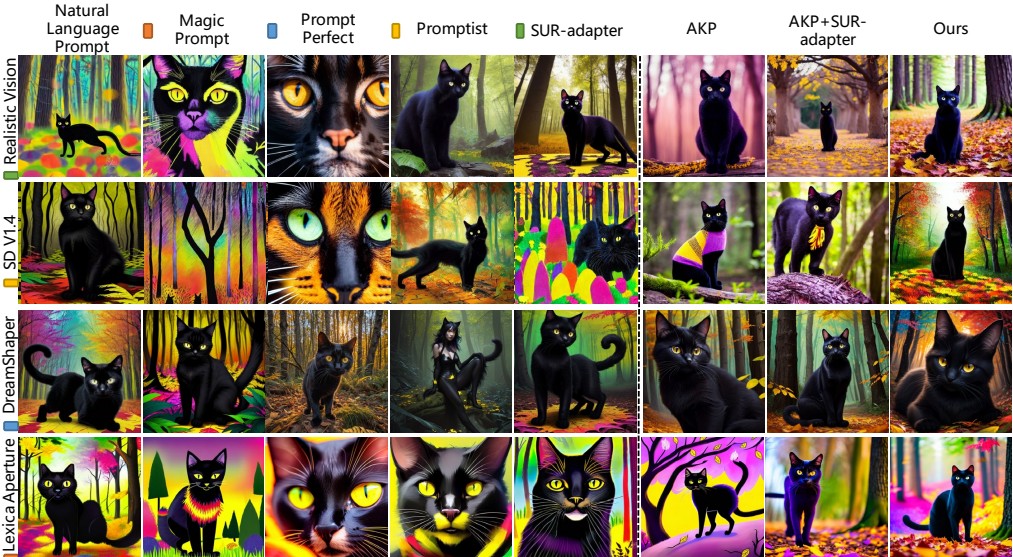

**Natural Language Prompt**: a black cat with yellow eyes, standing in a vibrant and colorful forest. The background is filled with a variety of colors, creating a visually striking and artistic atmosphere.

Figure 3: Results of diverse APO methods on MGAPO task. Each column displays images from the same APO method, and each row features images generated by the same SD model. APO methods with matching color and SD model share the same domain, indicating that the training data for the APO model was generated by that specific SD model.

## 4.4 Comparison to Other Methods

We compared various prompt optimization methods mentioned in Section 2. MagicPrompt [7], a popular prompt generator for SD from Hugging Face, PromptPerfect [16], a commercial tool for prompt optimization, Promptist [8], and SUR-adapter [37]. Additionally, automatic keyword prompts and manual prompts are keyword based prompts generated by GPT-3.5 based on image descriptions and manually designed prompts in our dataset, respectively. Furthermore, we integrate our ICL-based prompt rewriting with the SUR-adapter for comparison in our experiments. Concretely, we augment the representation of automatic keyword prompts to the representation of natural language prompts from the SUR-adapter, using it as the generation condition.

Table 1 shows that our proposed method outperforms most prompt optimization methods in semantic consistency and image quality. The results indicate that our proposed method exhibits good generalization performance in prompt optimization tasks. The APO method based on text optimization performs well on metrics that focus solely on image quality, such as AES scores and HPS, but falls short in terms of semantic consistency. In contrast, the SUR-adapter is able to maintain semantic consistency across different models, but does not perform well in terms of image aesthetics. Our method, however, can ensure both semantic consistency and aesthetics on unseen models. This conclusion is further strengthened by human evaluation results (Figure 4), where we see that our method consistently achieves higher win-rates than the other APO strategies.

## 4.5 Case Study

In Figure 3, we showcase the generalization performance of different APO methods. The SD models in the figure are all out-of-domain models for our approach. It is evident that most APO methods produce satisfactory results within their training domains but struggle to generate good results on out-of-domain models. For instance, Promptist generated a "cat-human" image on the Dreamshaper model. PromptPerfect generates narrow-angled images on all of the out-of-domain models. In contrast, our method consistently produces accurate and aesthetically pleasing images across all out-of-domain models. We provide more comparative cases in Appendix C.3.

To analyze the content contained in the trained domain prototypes, we artificially adjusted the weights $\eta_s$ in Equation 8 during the inference phase. We selected domain prototypes obtained from models with two different styles and treated $P_a$ as a different linear combination of the two. This is equivalent to adding domain information in different proportions to the generation conditions. In the Figure4a,

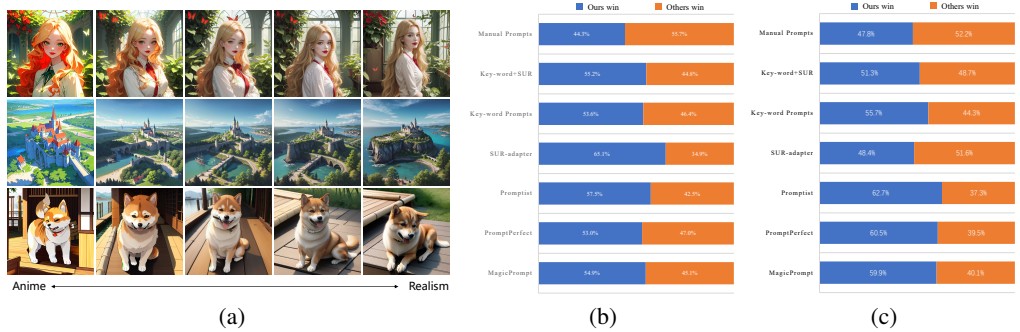

|  |  |  |  |
|---|---|---|---|
| Anime ← | | | → Realism |
| (a) | (b) | (c) |

Figure 4: (a) Linear combinations of domain prototypes from anime style models and domain prototypes from realism style models. The blending ratio changes from left to right. (b) Human Evaluation for Aesthetics. (c) Human Evaluation for Text-Image Alignment.

Table 2: Ablation study of the loss functions.

| $\mathcal{L}_r$ | $\mathcal{L}_d$ | $\mathcal{L}_a$ | $\mathcal{L}_c$ | BlipScore | AesScore | ImageReward |
|---|---|---|---|---|---|---|
| ✓ | | | | 0.303 | 6.110 | 0.254 |
| ✓ | ✓ | | | 0.313 | 6.139 | 0.258 |
| ✓ | ✓ | ✓ | | 0.324 | 6.275 | 0.384 |
| ✓ | ✓ | ✓ | ✓ | **0.332** | **6.384** | **0.427** |

Table 3: Effectiveness of domain prototypes.

|  | Color | BlipScore | AesScore | ImageReward |
|---|---|---|---|---|
| **Ours** | **0.477** | **0.332** | **6.384** | **0.427** |
| w/ random prototypes | 0.462 | 0.328 | 6.183 | 0.373 |
| w/o prototypes | 0.451 | 0.302 | 6.140 | 0.258 |

we show the results of mixing animation style domain and the realistic style domain. From left to right, we gradually reduce the weight of anime and increase the weight of realism. In the Appendix C.4, we provide more cases for further analysis of domain prototypes.

## 4.6 Ablation Study

**Effectiveness of Domain Prototypes.** To evaluate the effectiveness of domain prototypes, we employed three settings. The first setting involves not using domain prototypes, only utilizing the adapter for fine-tuning. The second setting involves using a fixed-parameter fully connected layer to align the dimensions of image features and domain prototypes, which is equivalent to randomly selecting image feature channels. The third setting is the one where we drop domain-insensitive channel information. The evaluation results are presented in the Table 3, showing that dropping domain-insensitive information effectively enhances the quality of the generated images.

**Effectiveness of Loss Functions.** We conducted ablation experiments on the four loss functions we adopted. The results are shown in the Table 2. The removal of the domain style distillation loss $\mathcal{L}_a$ led to a significant decrease in the quality of the generated images. The denoising loss $\mathcal{L}_d$ had a relatively minor impact on the generated results. Ultimately, the optimal performance emerged through the combination of all four losses, affirming the synergistic enhancement each proposed module provides. This underscores the significance of an integrated approach for superior model generalization, with each module playing a pivotal role. We visualize the ablation of loss functions in the Appendix C.5.

## 4.7 Visualization of Conditioned Features

We visualize the conditioned features used for text-to-image generation using the t-SNE algorithm. These include the original natural language descriptions of the images, keyword prompts optimized by the first-stage large language model, features output by the second-stage AP-adapter, and manually designed prompts.

We use different colors to represent different domains, as shown in Figure 5, reflecting the domain distinctiveness of various features. Both the natural language descriptions and keyword prompts have poor domain distinctiveness. We observe that the domain distinctiveness of the AP-adapter is even better than manually designed prompts. This is likely because we introduce image features that better represent domain differences. Therefore, the output of the AP-adapter performs better in domain distinctiveness than purely textual manual prompts.

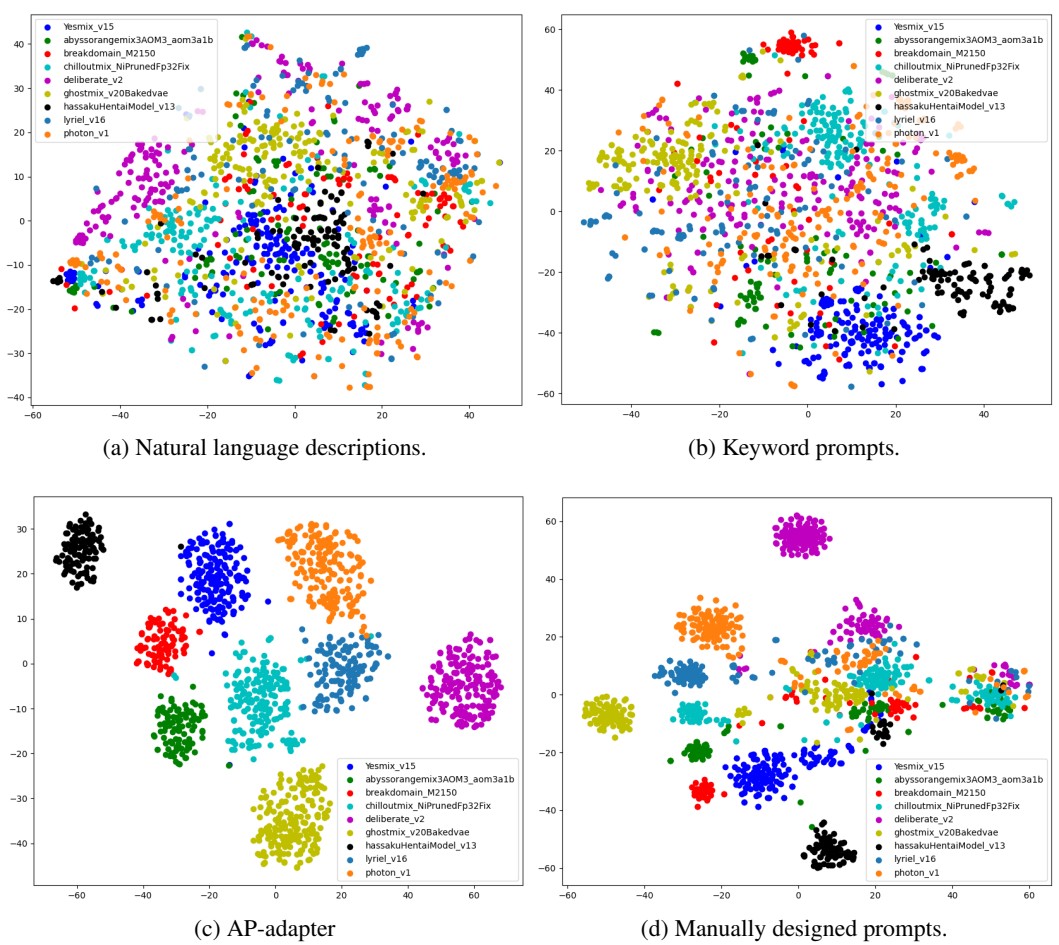

Figure 5: Visualization of text-conditioned domain distinctiveness. (a) Natural language description of the image. (b) Keyword prompts output by the first-stage large language model. (c) Features output by the second-stage AP-adapter. (d) Manually designed prompts.

## 5 Conclusion

In this paper, we propose a new task, MGAPO, which identifies shortcomings of previous APO methods in generalization of unseen models. To address this challenge, we propose a two-stage approach called AP-Adapter that combines both text optimization and representation optimization methods. In the first stage, we leveraged the zero-shot reasoning capability of large language models to rewrite natural language prompts to keyword prompts. In the second stage, we introduce the domain prototypes, which decodes images to obtain representation for various domains. These prototypes are then combined with text representations to align with the representations of manually designed prompts. Rigorous experimentation and evaluation affirm the effectiveness of our proposed approach in optimizing prompts for out-of-domain models, achieving superior results without compromising the quality of the generated images.

## Limitations

While our dataset offers valuable insights into the CIVITAI community, its relatively small size and focus on single-character images may limit the model's performance in handling more intricate and complex scenes. The necessity for an increasing number of domain prototypes as the dataset expands poses a challenge, leading to a growth in model parameters. Future work could explore leveraging parameter disparities among different models to tailor the APO optimization strategy, addressing this potential scalability concern.

## Acknowledgement

This work is supported by the National Natural Science Foundation of China under Grants Nos. 61972192, 62172208, 61906085. This work is partially supported by Collaborative Innovation Center of Novel Software Technology and Industrialization. This work is supported by the Fundamental Research Funds for the Central Universities under Grant No. 14380001.

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

# A Supplemental Algorithm Information

We illustrate our detailed training process in the Algorithm 1.

---

**Algorithm 1** Training pipeline

---

**Require:** Checkpoints $\mathcal{S} = \{\Theta_1, \Theta_2, ..., \Theta_K\}$, datasets $(n_i^k, m_i^k, I_i^k)_{i=1}^{N_k}$ in the source domain, $f_{clip}^T$ and $f_{den}$ of the stable diffusion model, $f_{clip}^I$ image encoder, $f_{llm}$ large language model, domain prototypes $\{P_k^{dom}\}_{k=1}^K (\phi_1)$, adapter $g_{ada}(\cdot; \phi_2)$.

**Ensure:** Trainable domain prompt parameters $\phi_1$ and adapter parameters $\phi_2$.

1: **Initialize:** Set $f_{llm}$ and $f_{clip}^I$ to pretrained parameters.
2: **for** T in training epochs **do**
3:     **if** T is even **then**
4:         Freeze domain prototypes parameters $\phi_1$.
5:     **else**
6:         Freeze adapter parameters $\phi_2$.
7:     **end if**
8:     **for** dataset of domain $i$ in datasets **do**
9:         Set $f_{clip}^T$ and $f_{den}$ to $\Theta_k$ parameters.
10:        Get keyword prompt token embedding $f_{clip}^T(f_{llm}(n_i^k))$ and manual prompt token embedding $(f_{clip}^T(m_i^k))$.
11:        Get domain prototypes $P_a$ by Equation 8.
12:        Concatenate domain prototypes and token embedding representations as adapter input.
13:        Obtain the output of the adapter $g_{ada}$.
14:        **if** T is even **then**
15:            Calculate values of loss functions by Equation 13.
16:        **else**
17:            Calculate values of loss functions by Equation 12.
18:        **end if**
19:        Update all learnable parameters.
20:    **end for**
21: **end for**

---

# B Supplemental Dataset Information

## B.1 Dataset Analysis

**Example of Data.** Figure 6 shows a sample from our dataset. We utilize the LLaVA-v1.5 13B model to generate natural language descriptions based on images. According to the query in the Figure 6, LLaVa generates detailed descriptions about the image content and style, typically consisting of 5 to 6 sentences. To filter out the most relevant parts with the image, we calculate the CLIP similarity between each sentence and the image, selecting the top 3 sentences with the highest similarity as natural language prompts.

**Example of Stable Diffusion Models.** In Figure 7, we list some of the stable diffusion (SD) models we used. The models are ranked based on the number of samples they contribute to the dataset, with the top 20 models displayed. This ranking also reflects the popularity and generation quality of these models within the community. These models mainly come from two sources: one is models obtained by fine-tuning SD with custom data, and the other is models obtained by merging the weights of other models.

**Distribution of Sample Size.** As shown in the Figure 8, we present the distribution of samples for all models in the dataset. For clarity, we use a sample quantity of 35 as the threshold. It can be observed from the graph that the number of models with a large number of samples is relatively small, with many models having fewer than 10 samples. Therefore, we divide our dataset into three parts, as mentioned in Section 4.1. As stated, we assign models with a higher number of samples to the training and test sets, and the remaining samples were used as the demonstration library for Section 3.3.

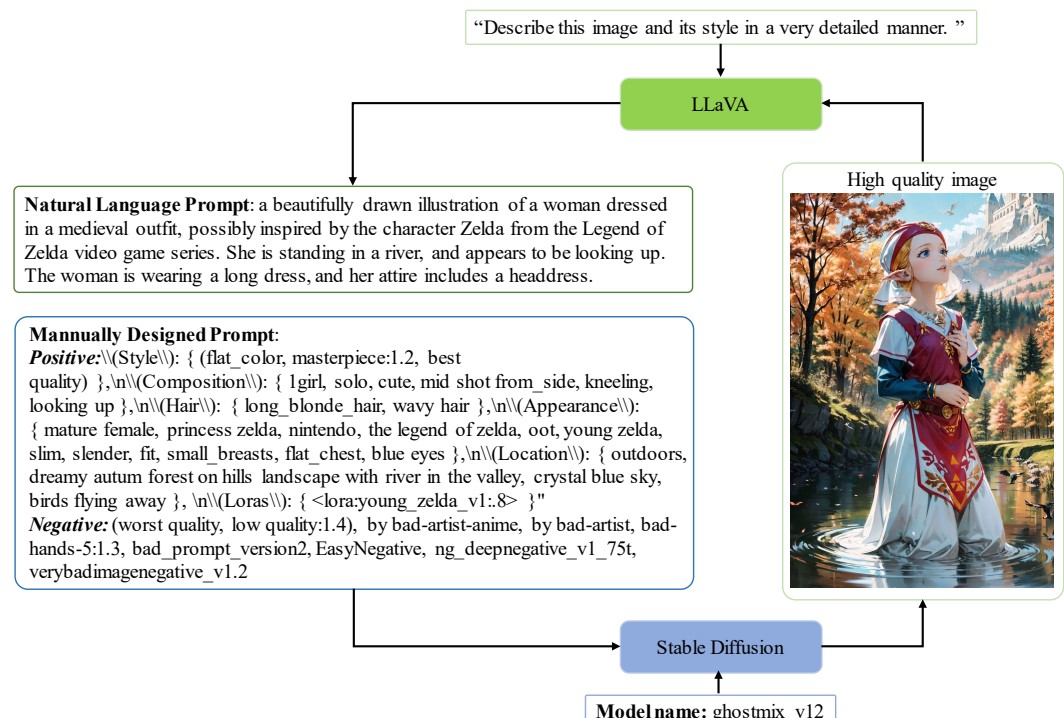

Figure 6: A sample from the dataset, each containing a natural language prompt generated by LLaVa, a manually designed prompt, a high-quality image, and model information.

## B.2 Ethical Statement

Due to the nature of our dataset, which is entirely generated by diffusion models, it is unlikely to be used for systems that may violate personal privacy. Additionally, we have filtered out samples containing explicit and nude images from the data. Upon careful examination of our dataset, we believe it is unlikely to provide harmful information.

## C  Supplemental Experiments

### C.1  More Implementation Details

In the ICL-Based Prompt Rewriting stage, we utilize the gpt-3.5-turbo API to generate automatic keyword prompts. It is noteworthy that, at this stage, the large language model generates automatic keyword prompts based on the demonstrations provided by ICL, including positive and negative prompts. Only positive prompts participate in training, while negative prompts serve as negative conditions for image generation during the testing phase. Here we provide the automatic keyword prompts generated from the natural language prompts in Figure 3.

Positive Prompts: masterpiece, best quality, vibrant colors, artistic atmosphere, black cat, yellow eyes, forest scene, visually striking, colorful background, unique appearance, eye-catching, animal focus, detailed, high resolution, solo, nature theme, visually appealing, beautiful, gorgeous

Negative Prompts: lowres, bad anatomy, bad eyes, error, missing features, distorted features, worst quality, low quality, normal quality, jpeg artifacts, signature, watermark, username, blurry, bad pictures, disconnected elements, unnatural colors, bad composition, overexposure, underexposure, monochrome, bad prompt, horror, easynegative.

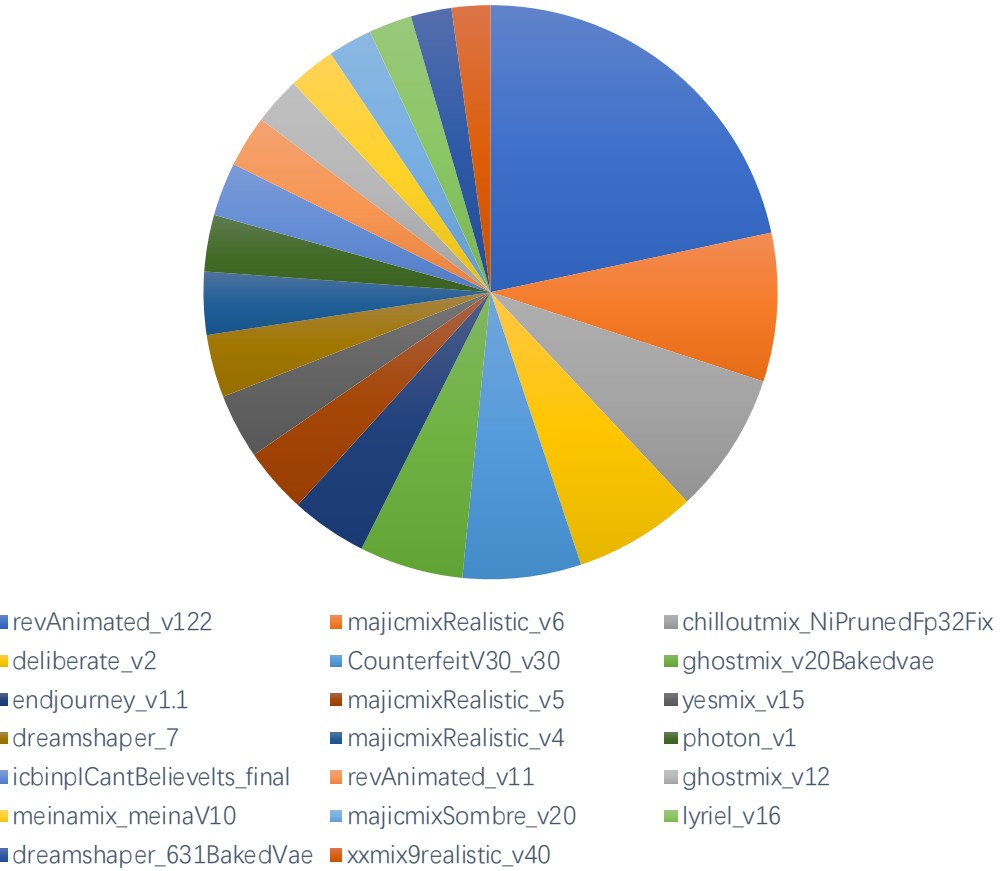

revAnimated_v122 ■ majicmixRealistic_v6 ■ chilloutmix_NiPrunedFp32Fix
deliberate_v2 ■ CounterfeitV30_v30 ■ ghostmix_v20Bakedvae
endjourney_v1.1 ■ majicmixRealistic_v5 ■ yesmix_v15
dreamshaper_7 ■ majicmixRealistic_v4 ■ photon_v1
icbinpICantBelieveIts_final ■ revAnimated_v11 ■ ghostmix_v12
meinamix_meinaV10 ■ majicmixSombre_v20 ■ lyriel_v16
dreamshaper_631BakedVae ■ xxmix9realistic_v40

Figure 7: The top 20 models by sample size, along with the respective proportions.

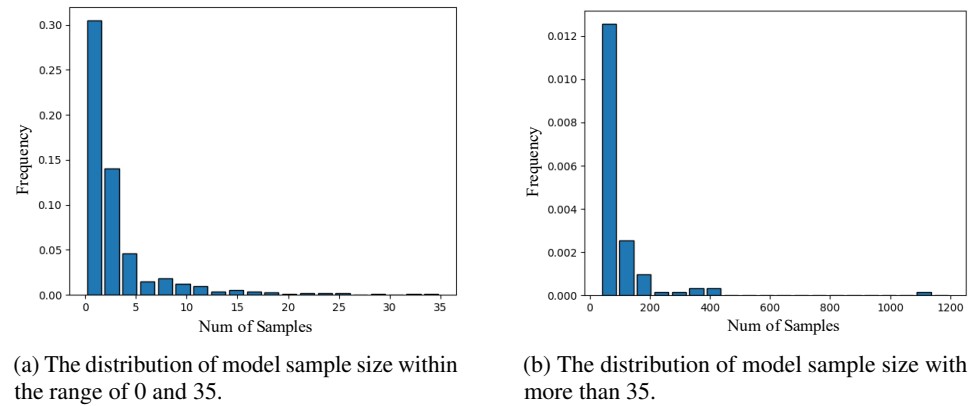

(a) The distribution of model sample size within the range of 0 and 35.

(b) The distribution of model sample size with more than 35.

Figure 8: The distribution of model sample size.

In the Prototype-Based Prompt Adaptation stage, all models are trained on two NVIDIA RTX 3090 GPUs, with steps set to 10000, batch size set to 16, and image resolution set to 512. Standard data augmentation techniques, such as normalization, resizing, and horizontal flipping, are applied during training. As for the platform to implement our network, we use PyTorch 2.1.

### C.2   Module Analysis

**Different CLIP Image Encoder Layers for Domain Discrimination.** To effectively acquire domain information, we tested the performance of the outputs of all layers in CLIP ViT-L/14 for domain

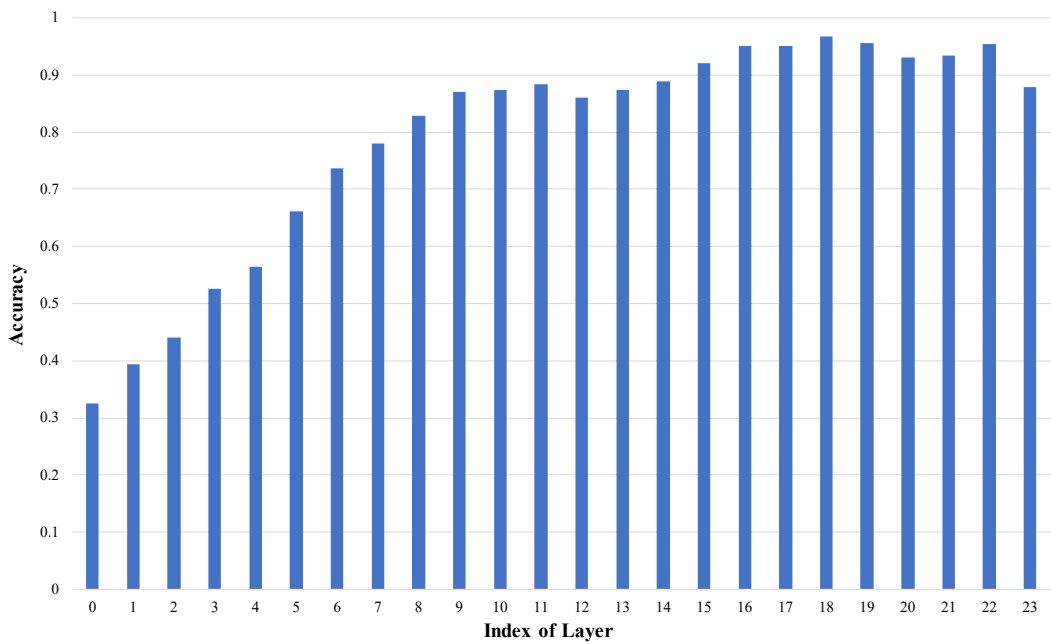

Figure 9: Domain discrimination accuracy at different layers of the CLIP image encoder.

classification, as shown in the Figure 9. We found that as the layer index increases, the domain classification ability of image embeddings is not strictly monotonically increasing. This phenomenon may be due to domain information being more related to image style rather than semantic information. As the layers of the CLIP image encoder deepen, the output embeddings contain more semantic information.

**Different backbone of the CLIP image encoder.**The evaluation results show the impact of different CLIP image encoders on the domain discriminator and the final generated results. Table 4 indicates that the choice of backbone has a minor influence on both domain classification and the quality of generated images. Additionally, it is observed that, across all backbones, features from deeper layers tend to be more advantageous for domain discrimination.

Table 4: Evaluation results of the impact of different backbones of the CLIP image encoder on the generated images. Acc refers to the classification accuracy of the domain discriminator.

|  | Acc | BlipScore | AesScore |
|---|---|---|---|
| **RN50** | 0.925 | 0.304 | 6.282 |
| **RN101** | 0.954 | 0.318 | 6.351 |
| **ViT-B/32** | 0.942 | 0.311 | 6.302 |
| **ViT-L/14** | 0.960 | 0.332 | 6.384 |

## C.3 More Cases for Comparison

We present more image generation results on out-of-domain models in Figure 10 and Figure 11. The models in Figure 10 follow the setup from Figure 3, while Figure 11 provides additional results from various models.

According to the results, it can be observed that images generated using our method perform well in both semantic consistency and aesthetic quality across all models. In contrast, results generated by other methods are not stable across different models.

## C.4 More Cases for Domain Prototypes Analysis

We visualized the impact of domain prototypes on image generation in Figure 4a, Figure 12, and Figure 13. In Figure 4a and Figure 12, we calculate $P_a$ using the following formula:

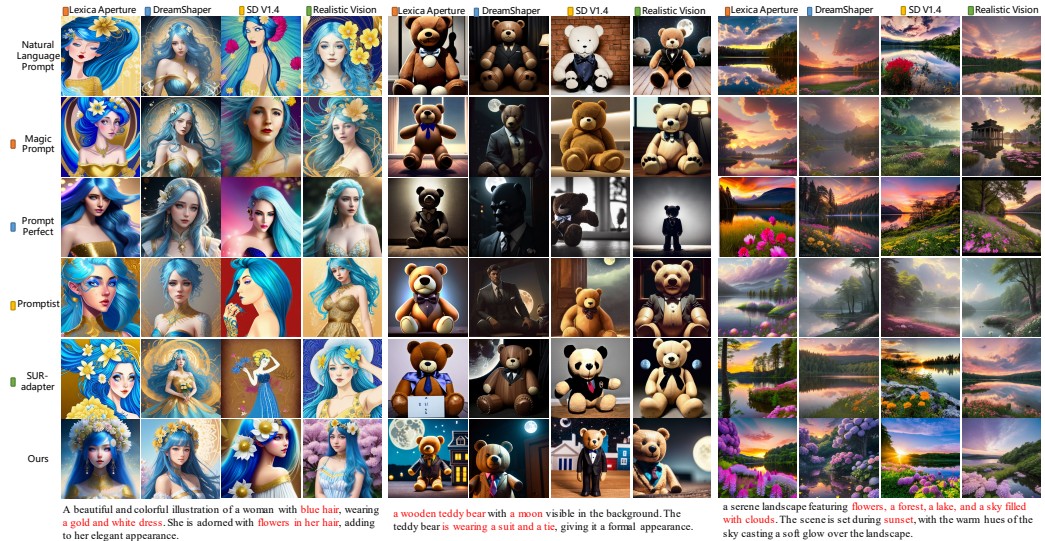

Figure 10: More comparative results of generated images on both in-domain and out-of-domain models.

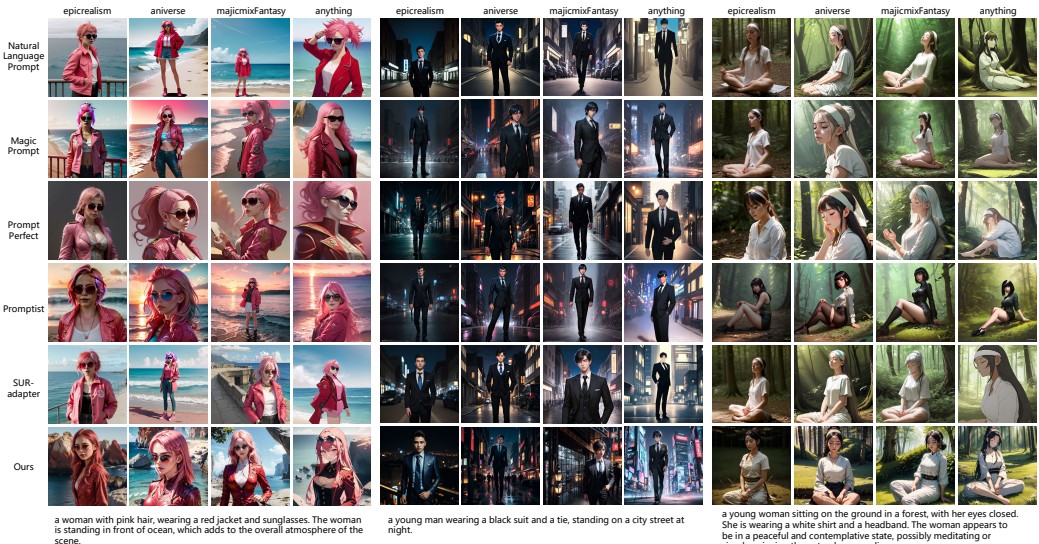

Figure 11: More comparative results of generated images on out-of-domain models.

$$P_a = (1 - \eta)P_i^{dom} + \eta P_j^{dom} \tag{14}$$

which $\eta \in [0, 1]$.

In these cases, it is evident that domain prototypes possess clear semantic information, representing specific image styles such as cartoon, realistic, or 3D animation.

However, domain prototypes do not always possess clear semantic information. As shown in Figure 13, we gradually inject domain prototypes into the conditional information by adjusting the $\eta_i$ value of $P_i^{dom}$. It seems difficult to define the semantics represented by the domain prototypes in these cases. We believe this reflects a comprehensive factors such as image detail richness, photography techniques, lighting, image contrast, and image softness.

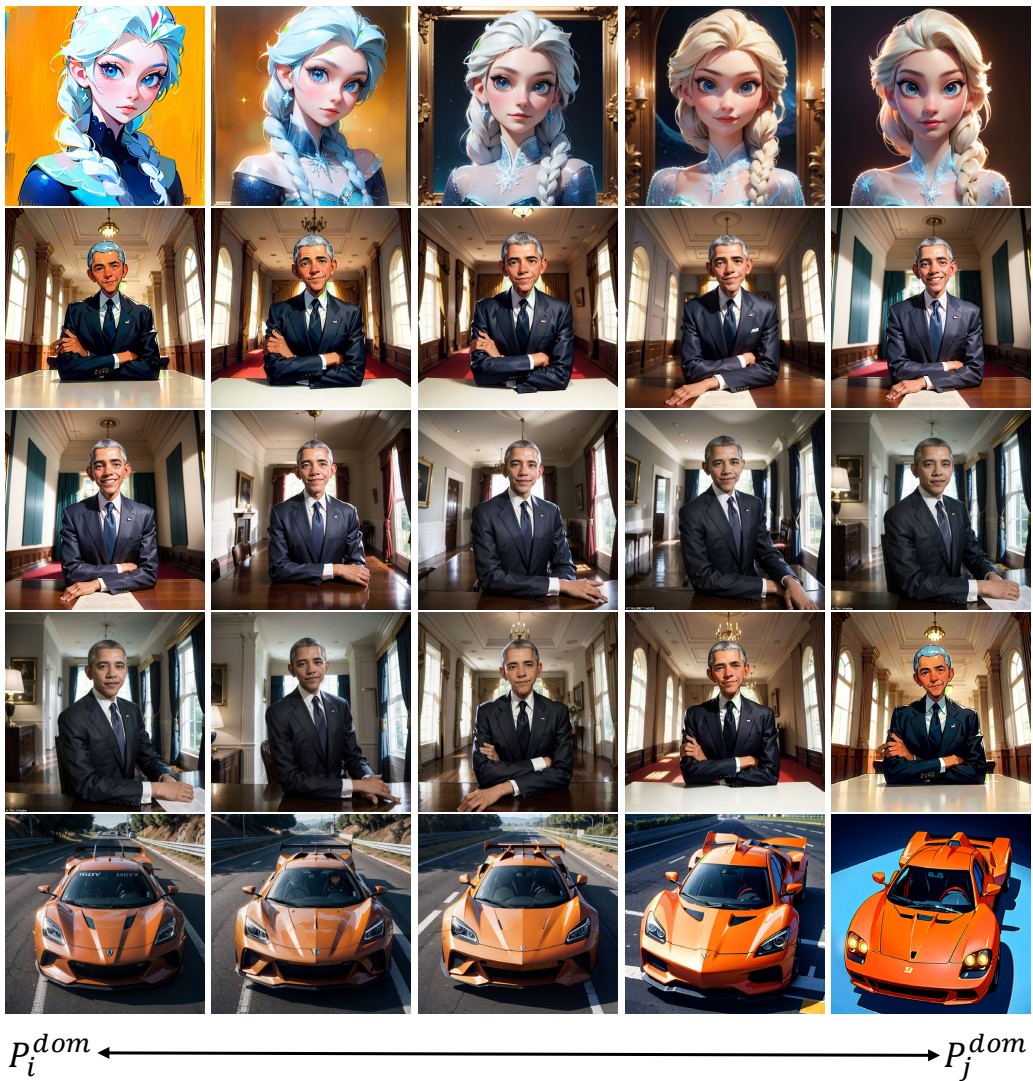

$P_i^{dom}$ ⟵———————————————————⟶ $P_j^{dom}$

Figure 12: Linear combinations of domain prototypes from $i$th models and domain prototypes from $j$th models.

## C.5 More Ablation Study

In Figure 14, we present additional results from the ablation experiments on the loss functions. Sequential ablation of each of the four loss functions reveals that the absence of any single loss function variably affects the image quality.

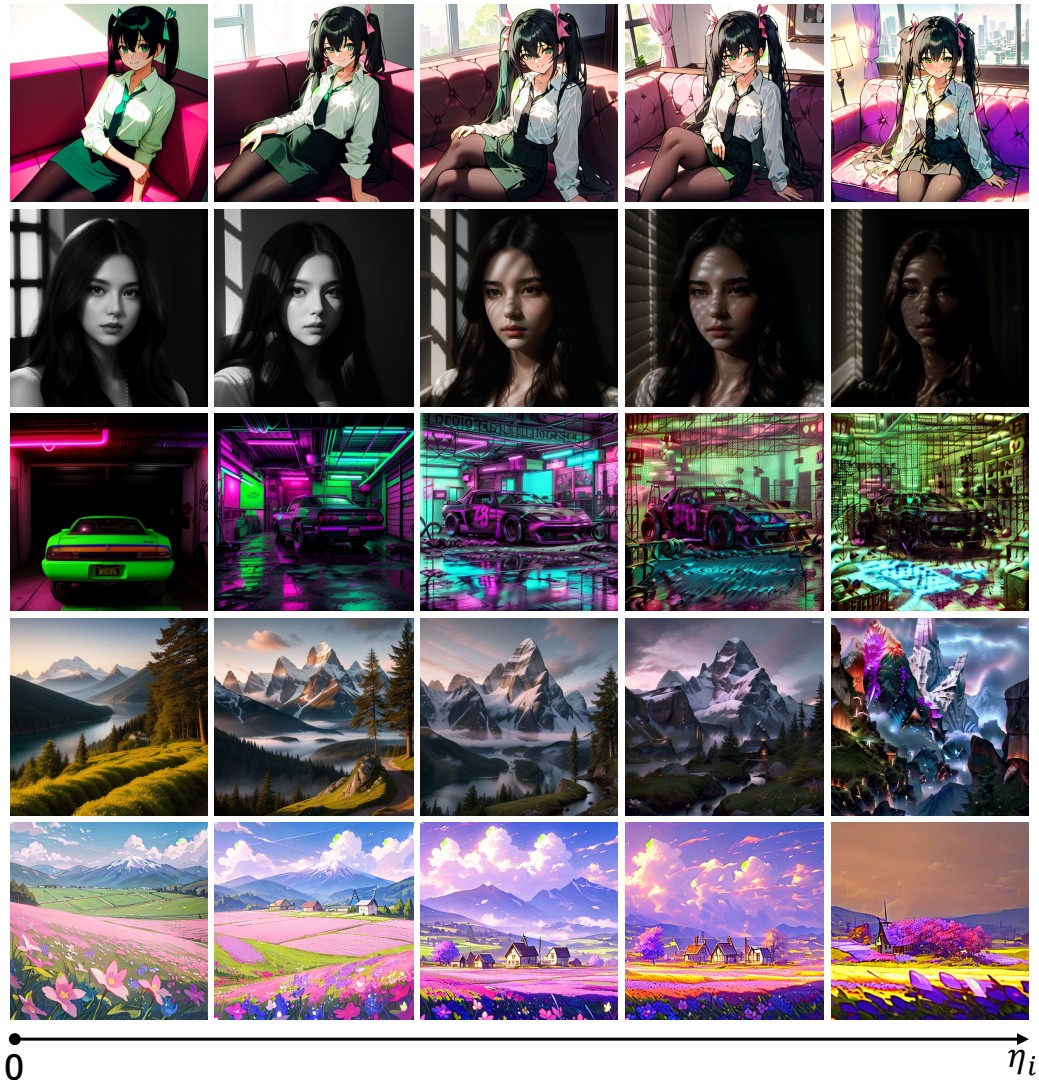

$\eta_i$

0

Figure 13: Injection of $i$th domain prototype information by adjusting the value of $\eta_i$.

| w/o $\mathcal{L}_c$ | w/o $\mathcal{L}_a$ | w/o $\mathcal{L}_r$ | w/o $\mathcal{L}_d$ | w/ all $\mathcal{L}$ |

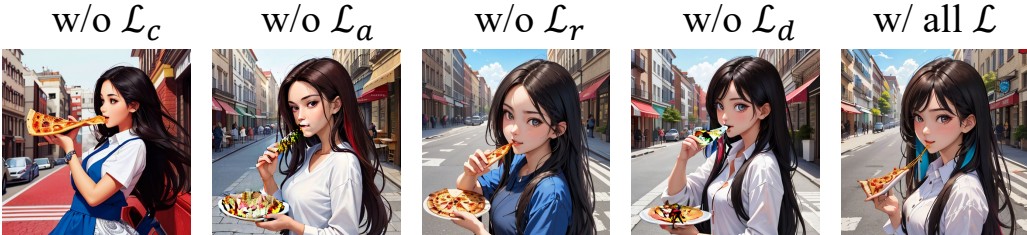

A woman with long, dark hair, wearing a white shirt, and eating a slice of pizza. She is posing for the camera while enjoying her meal. The scene takes place on a street.

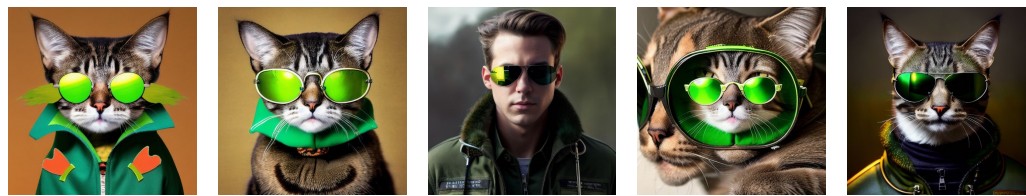

A cat wearing a green jacket and sunglasses, giving it a stylish and unique appearance. The cat is positioned in the center of the image.

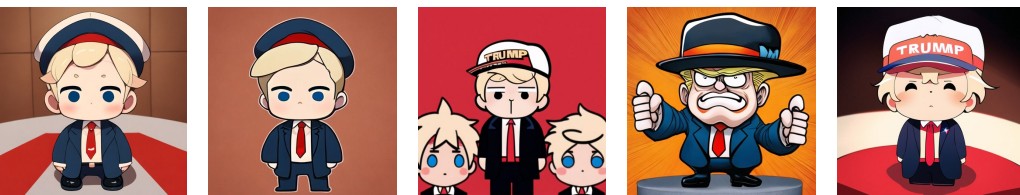

A cartoon figure of a man wearing a suit and a hat. The man appears to be a caricature of Donald Trump, the former U.S. president.

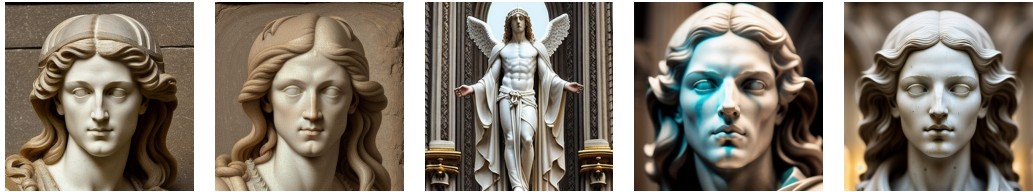

A plaster portrait from the Renaissance period. The plaster appears to be a man with long hair, facing the camera directly.

Figure 14: Demonstration of Ablation Study of Loss Functions.

