# OpenReview forum: "AP-Adapter: Improving Generalization of Automatic Prompts on Unseen Text-to-Image Diffusion Models"
_NeurIPS.cc/2024/Conference — NeurIPS 2024 poster_

### Official Review · Reviewer_fgUM · 2024-07-07

**Soundness:** 2
**Presentation:** 3
**Contribution:** 3
**Rating:** 6
**Confidence:** 4

**Summary:**

This paper proposes a new task called MGAPO, aimed at addressing the generalization problem of existing APO methods on unseen text-to-image generation models. To achieve this, the authors present a two-stage method called AP-Adapter. In the first stage, keyword prompts are generated through a large language model; in the second stage, an enhanced representation space is constructed by leveraging inter-model differences, and prompt representations are adjusted through domain prototypes to enable generalization to unseen models. Experimental results demonstrate that AP-Adapter can generate high-quality images on previously unseen diffusion models and outperform existing methods in both semantic consistency and aesthetic quality.

**Strengths:**

1 The introduction of the MGAPO task makes text-to-image model design more applicable to real-world scenarios.

2 The related concepts and differences are clearly articulated, especially in Figure 1.

**Weaknesses:**

1 The core of this paper lies in the computation of domain prototypes, which are influenced by CLIP. However, the authors did not conduct relevant ablation experiments.

2 Figure 4 includes human evaluation, which introduces a degree of subjectivity. Additionally, previous methods like sub-adapter did not conduct similar validation. Furthermore, the authors seem to lack detailed explanations regarding human validation in this paper.

3 The proposed method is relatively complex, introducing additional models such as a pre-trained LLM and CLIP for prototype computation, which increases overall complexity and training/inference time. The authors did not discuss these aspects.

4 AP-Adapter still requires a large number of manually designed prompts as references during training, which may increase the workload and complexity of data preparation.

**Questions:**

see Weaknesses

**Limitations:**

The authors mentioned in the Limitation section that the main drawback of this paper is the small size of the dataset used, which is focused on single-character images. This limits the model's performance. Furthermore, the authors explained that as the dataset size expands, the number of domain prototypes might increase.

---

> ### Author Rebuttal · Authors · 2024-08-07
>
> Thank you for the constructive comments. We will address the concerns below.
>
>
> Q1. The authors did not conduct relevant ablation experiments on CLIP.
> A1. The computation of domain prototypes indeed relies on the image encoder. Moreover, domain prototypes are concatenated with text representations to serve as control conditions for image generation, so they should be as similar to text representations as possible. Given that stable diffusion 1.5 uses CLIP as the text encoder, we also use CLIP as the image encoder, ensuring that the image representations share a feature space with the text representations.
> In the appendix, we explore the impact of CLIP on domain prototypes. The experiments include Figure 8 and Table 4 in section C.2.
>
>
> Q2. The human validation lack detailed explanations.
> A2. We followed the approach presented in the SUR-adapter[1]. We collected 108 valid questionnaires. Participants were shown images generated by our method and baseline methods, along with their corresponding descriptions. They were asked to choose the better image based on the questions "Which image do you think has higher quality?" and "Which image do you think better matches the text description?" We then compiled and analyzed the survey results.
>
>
> Q3. The proposed method introduces additional models such as a pre-trained LLM and CLIP for prototype computation.
> A3. The LLM is only used in the first stage for automatic prompt generation, employing the ICL strategy, and does not participate in subsequent training. The LLM’s model parameters are not updated. During training, the parameters of CLIP are frozen and do not get updated. During inference, CLIP is no longer used; only the trained domain prototypes are involved. Therefore, training and inference times are not significantly increased.
>
>
> Q4. The data collection process adds to the workload and complexity of the proposed method.
> A4. All existing APO methods require a large number of manually designed prompts as references for training.
> Moreover, such data collection and retraining should be repeated for the APO on each newly-coming diffusion model.
> Our method is desined to address this issue by eliminating the need for new data preparation and retraining when encountering new diffusion models. Our trained adapter can be directly used on new models without retraining.
>
>
> [1] Zhong S, Huang Z, Wen W, et al. Sur-adapter: Enhancing text-to-image pre-trained diffusion models with large language models[C]//Proceedings of the 31st ACM International Conference on Multimedia. 2023: 567-578.

---

> > ### Comment · Reviewer_fgUM · 2024-08-13
> >
> > Thank you for your response. It addressed some of my concerns, and I am willing to increase my confidence level.

---

> > > ### Author Response · Authors · 2024-08-13
> > >
> > > Dear Reviewer,
> > >
> > > We greatly appreciate your detailed review and insightful comments.

---

### Official Review · Reviewer_W76C · 2024-07-09

**Soundness:** 2
**Presentation:** 2
**Contribution:** 2
**Rating:** 6
**Confidence:** 4

**Summary:**

The paper proposes model-generalized automatic prompt optimization (MGAPO), an automatic prompt optimization (APO) method which trains on a set of known models to enable generalization to unseen models during testing. MGAPO presents significant challenges, a perspective missing in previous methods. MGAPO includes a two-stage prompt optimization method. In the first stage, a LLM is used to rewrite the prompts and generate keyword prompts. In the second stage, the methods leverage inter-model differences (derived from varied checkpoints) to captures the characteristics of multiple domains and store them as domain prototypes. These prototypes serve as anchors to adjust prompt representations, enabling generalization to unseen models. The optimized prompt representations are subsequently used to generate conditional representations for controllable image generation.

**Strengths:**

- The paper presents a new perspective to the automatic prompt optimization at the feature level.
- Authors have shown the effect of each loss component in Table 2.

**Weaknesses:**

- I believe the quality of writing can be improved. The language of paper seems unnecessarily complex and hard to comprehend at certain places especially the methodology section.
- The problem solved in the paper seems more of an incremental learning problem than domain generalization. The training and target domains differ only in terms of number of images seen. They do not differ in term of style or content as per paper. What are the views of authors about this?
- I can see in figure 2 that the UNET of diffusion model is freezed. If that's the case, then it renders the diffusion loss $\mathcal{L}_d$ in equation 11 useless. What exactly is being updated in equation 11? The $\theta$ parameters of UNET are freezed.
- In Fig 3 (row 3), the proposed method doesn't seem to produce optimal results with DreamShaper. The cat is supposed to be standing as per the prompt, however, the cat generated by the proposed method seems to be sitting. Compared to proposed method, the baselines seems to be working better in this case.
- Fig. 2 seems to be hastily drawn. There are many things which can improved in the figure (minor: "M" of LLM block is going out of the box). Also the font of in-context learning block is very small
- The gains in Table 1 seem to be average compared to baselines. In fact baselines are better in certain cases. Given the complexity of the proposed method, the gains are expected to be larger, which is not the case.

Overall, the authors are encouraged to answer the above queries and provide solid explanations

**Questions:**

Please check the weaknesses section.

**Limitations:**

Authors have provided the limitations statement.

---

> ### Author Rebuttal · Authors · 2024-08-07
>
> Thank you for the constructive comments. We will address the concerns below.
>
> Q1. The quality of writing can be improved.
> A1. Thank you for your suggestions. We will optimize the writing in the methodology section to make it more concise and easy to understand.
>
>
> Q2. What’s the difference between the training and target domains?
> A2. In our dataset, each checkpoint is fine-tuned using images of different styles and content, resulting in varying generative capabilities. During experiments, we use a subset of these checkpoints as the training set and test on another subset. The fundamental difference between source and target domain checkpoints lies in the style and content of the fine-tuning data, not the number of the data.
>
>
> Q3. What exactly is being updated in equation 11?
> A3. In Equation 11, the parameters of $ q_i^k $ will be updated. $ q_i^k $ is the output of $ g_{\text{ada}} $ in Equation 9. Thus, the variables being updated are the parameters of the transformer layers and linear layers in the adapter.
>
>
> Q4. Compared to proposed method, the baselines seems to be working better in Figure 2.
> A4. We found that the issue lies in the ICL-based prompt rewriting in the first stage. As shown in Appendix C.1, the prompts rewritten by the large language model do not include the word "standing." This indicates that while the adapter in the second stage can enhance the generalization of the automatic prompts to new models, it cannot compensate for the information lost by LLM in the first stage. Therefore, the quality of the automatic prompts is also crucial.
>
>
> Q5. Fig. 2 need to be improved.
> A5. Thank you for your advice. We will improve the quality of the figures and provide clearer figures in the final version.
>
>
> Q6. Given the complexity of the proposed method, the gains are expected to be larger, which is not the case.
> A6. Compared to baseline methods, our approach requires significantly fewer parameters to be trained, limited only to the domain prototypes and adapter. Therefore, our method is not more complex. Additionally, our approach performs better than other methods in most cases, as shown in Table 1 and Figure 2.

---

> ### Comment · Reviewer_W76C · 2024-08-08
>
> Thank you for your response. Based on the response of the authors, I have few more queries below:
>
> - Q2: Authors mention that images sourced until checkpoint 40 are taken as source data and from checkpoint 41 as target data and they differ in content and style. How do authors ensure the style and content of target images are different. They are still from the same data distribution, hence more clarity is needed at this point.
>
> - Q4:  Authors argue that LLMs inability to generate the correct pose for cat is the reason behind the incorrect pose of cat in the final image. This suggests that the proposed method is heavily dependent on the quality of the prompt generated by the LLM. How do authors aim to improve or get rid of this dependency? or is there any alternate way to get around such issues
>
> - Q6: I think authors want to point out at Fig.3 (Fig 2 is a block diagram). Kindly request the authors to put correct references. The scores are still marginal. I do not see any steep jump in scores of semantic consistency. For example proposed method has a color score of 0.477 while SUR adapter has 0.472 which is very marginal increment. Same is with other cases as well.

---

> > ### Author Response · Authors · 2024-08-09
> >
> > We appreciate your reply. We will address the concerns below.
> >
> >
> > Q2. How do authors ensure the style and content of target images are different?
> > A2. In the dataset we collected, each image is annotated with the source checkpoint that generated it and the manually designed prompt used for its generation. The style and content of the image are determined by these two factors: the source checkpoint and the prompt, respectively.
> > Regarding style, each checkpoint is uploaded to Civitai.com by users and is typically fine-tuned with users’ private data or data of particular interest. As a result, the fine-tuning data for different checkpoints can be viewed as coming from different data distributions. This variation in fine-tuning data leads to differing generative capabilities across checkpoints, resulting in differences in the styles of the images they generate.
> > Regarding content, the prompt-image pairs we collected also come from different checkpoints, and these prompts are usually designed by users to highlight the generative capabilities of their respective checkpoints in specific domains. Figure 14 in the Appendix illustrates the differences among prompts, demonstrating that prompts from different checkpoints exhibit significant differences in content. As shown in Figure 14(d), different colors represent prompts from different checkpoints, and we can observe that the same colors tend to cluster together, indicating a clear semantic distinction between the prompts of different checkpoints.
> > Therefore, in our dataset, the images generated from different checkpoints typically differ in both style and content and cannot be considered as coming from the same data distribution.
> >
> >
> > Q4. How do authors aim to improve or get rid of the dependency of the quality of automatic prompt? or is there any alternate way to get around such issues.
> > A4. Our work primarily focuses on improving the generalization capability of automatic prompts across different checkpoints, with less emphasis on their generation. For the generation of automatic prompts, we only simply use a LLM without fine-tuning. To address the dependency of the quality of the automatic prompts, we can consider it from two perspectives:
> > 1. Use the natural language prompt as an extra input of the adapter. In this way, by introducing some specially-designed modules into adapter, the adapter can have additional capabilities for semantic completion of automatic prompts, thereby reducing its reliance on the first stage.
> > 2. Develop a more effective automatic prompt generation module. By jointly training this module with the adapter, we can enhance the module's ability to capture semantics, ensuring that the generated prompts have higher semantic quality while better aligning with the generalization requirements of the adapter.
> >
> > Q6. In terms of semantic consistency, the improvement of the method in this paper over SUR-adapter is modest.
> > A6. We apologize for the earlier misreference; the correct figure is Figure 3.
> >     Among the baselines, SUR-adapter mainly focuses on enhancing semantic consistency between prompts and images on a specific diffusion model by using an adapter. Our proposed method also employs an adapter but focuses on improving its generalization capability. Consequently, although our method offers only modest gains in semantic consistency compared to the SUR-adapter, it achieves a substantial improvement in image quality, as demonstrated in Table 1.
> >    Among other baselines, PromptPerfect focuses on improving image quality and achieves an Aesthetic Score of 6.249, which is close to our score of 6.384. However, our method significantly outperforms PromptPerfect in semantic consistency.
> >    In summary, our method achieves strong performance in both semantic consistency and image quality, whereas previous methods did not achieve both.

---

> > > ### Comment · Reviewer_W76C · 2024-08-09
> > >
> > > Thanks to authors for providing a detailed response to my queries. I am satisfied with most of the answers. However, regarding Q2, the out-of-distribution nature of data samples is based on the authors' assumptions which may or may not be true. Overall, I would like to increase my score to weak accept.

---

> > > > ### Author Response · Authors · 2024-08-14
> > > >
> > > > Dear Reviewer:
> > > >
> > > > We sincerely appreciate your insightful comments and your willingness to increase the score.
> > > >
> > > > Regarding the issue of the "out-of-distribution nature of data samples" mentioned in Q2, we would like to provide some further clarification.
> > > >
> > > > Our assumption that "relevant data from different checkpoints may have different data distributions" is not merely a conjecture but is based on our observations and experiments. The "out-of-distribution nature of data samples" can be summarized from three aspects:
> > > >
> > > > 1. **The prompts from different checkpoints have different distributions.** As shown in Figure 14, we attempted to map the prompts collected from various checkpoints into the same feature space and found that they are clustered according to the checkpoints, exhibiting domain distinctiveness.
> > > >
> > > > 2. **The images generated from different checkpoints exhibit domain differences.** As shown in Figure 8, we attempted to classify the source checkpoints of images collected from various checkpoints. We found that using only a simple linear classifier with pretrained CLIP encoder achieved around 95% accuracy. This indicates that images generated from different checkpoints have domain distinctiveness.
> > > >
> > > > 3. **The checkpoints fine-tuned with different data have shifted model parameters and should not be considered the same model.** As shown in Figure 3, the third column illustrates our attempt to apply "Prompt Perfect", which is specifically trained for the "DreamShaper" checkpoint, directly to four different checkpoints: "DreamShaper", "Realistic Vision", "SD V1.4", and "Lexica Aperture". We find that applying "Prompt Perfect" to the latter three unseen checkpoints can sometimes lead to semantic loss and worse performance. Other columns also show that the performance of APO methods on unseen checkpoints is sometimes worse. This demonstrates that checkpoints also exhibit domain distinctiveness.

---

> ### Author Response · Authors · 2024-08-12
>
> Dear Reviewer,
>
> We sincerely appreciate your engagement in the discussion and your recognition of our rebuttal efforts. Thank you very much for your willingness to increase the score.
>
> Since the rating in the original review has not yet been updated, we are unsure whether this is an oversight and would like to kindly confirm if the rating has been changed in the system.
>
> Thank you once again for your time and valuable comments.

---

> ### Comment · Reviewer_W76C · 2024-08-14
>
> Thank you for further clarification. I appreciate authors putting more work on explaining the things. I am increasing my score to weak accept. The paper is good however, the increments as compared to the baselines are not that great.

---

### Official Review · Reviewer_redB · 2024-07-13

**Soundness:** 3
**Presentation:** 2
**Contribution:** 2
**Rating:** 5
**Confidence:** 3

**Summary:**

1. The authors propose model-generalized automatic prompt optimization (MGAPO), which targets the effectivenss of automatic prompts on unseen models.

2. The authors propose AP-Adapter which include in-context learning based prompt rewriting and prototype-based prompt adaptation.

3. The authors build a multi-modal, multi-domain dataset for training and evaluation.

**Strengths:**

1. The authors introduce the process of collecting and creating the multi-modal multi-domain dataset for training and evaluation.

2. The experimental results show that the proposed method outperforms existing baseline models.

3. The paper includes extensive further analyses and ablation studies to deepen the understanding of the proposed method.

**Weaknesses:**

1. While the proposed method outperforms other baselines, its increased performance appears quite marginal, particularly considering the significant gap compared to the performance of manual prompts. I am curious about the main reasons for this substantial gap between automatic and manual prompts, and what fundamental limitations of automatic prompting hinder its improvement. Are there any promising directions to overcome these limitations? Can we consider the proposed method as addressing some of these limitations?

2. The proposed prototype-based adaptation utilizes the concept of DomainDrop in a reverse manner. DomainDrop aims to learn domain-invariant features by eliminating domain-sensitive information. Conversely, the proposed method removes domain-insensitive information to construct domain prototypes. However, in the context of domain generalization, domain-insensitive information is often considered highly valuable, as it contains information shared across different domains, which is crucial for generalizing to unseen domains. Therefore, I am curious if the process of removing domain-insensitive information in the proposed method could result in the loss of important information. Additionally, relying on domain prototypes may lead to situations where a model cannot properly handle data from completely new domains that cannot be adequately matched with existing domains. In this case, we need to consider the difficulty of continuously increasing the number of domain prototypes, as this would require an increase in model size as well.

3. While using different checkpoints, the data used for training and testing are consistent in that they are all collected from CIVITAI and fine-tuned based on SD1.5. However, the pre-trained data used for other baseline methods does not necessarily share these characteristics. I am curious if this could have introduced some biases in favor of the proposed method, leading to better results.

**Questions:**

Please refer to the "Weaknesses" section.

**Limitations:**

This paper includes a specified section for limitations and the authors adequately addressed them.

---

> ### Author Rebuttal · Authors · 2024-08-07
>
> Thank you for the constructive comments. We will address the concerns below.
>
>
> Q1-1. What’s the the main reasons for this substantial gap between automatic and manual prompts?
> A1-1. The effectiveness of manual prompts lies in their iterative nature, allowing humans to refine prompts based on the generated images until the desired quality is achieved. In contrast, automatic prompts aim to replicate this effectiveness by learning patterns from manual prompts.
>
>
> Q1-2. What fundamental limitations of automatic prompting hinder its improvement?
> A1-2. The key limitation is that APO models struggle to evaluate whether an image meets specific requirements and to re-optimize the prompts accordingly.
>
>
> Q1-3. Are there any promising directions to overcome these limitations?
> A1-3. One potential solution is to incorporate reinforcement learning by designing a reward model that simulates human judgment. This would enable the model to dynamically identify and correct deficiencies in the current prompts based on the generated images.
>
>
> Q1-4. Can we consider the proposed method as addressing some of these limitations?
> A1-4. Our approach does not focus on generating superior automatic prompts but rather on enhancing the generalization of automatic prompts across different checkpoints, thereby avoiding the need for repetitive retraining of the prompt generation model for new checkpoints.
>
>
> Q2-1. If the process of removing domain-insensitive information in the proposed method could result in the loss of important information?
> A2-1. As shown in Figure 2, in the adapter section, the final adapted embedding is obtained by concatenating the prompt embedding and the Prototype-anchored embedding, followed by transformation. During this process, domain-insensitive information remains in the prompt embedding and is not completely removed, ensuring that important semantic information is not lost.
>
>
> Q2-2. Is it necessary to continuously increase the number of domain prototypes to adapt to new data?
> A2-2. We do not require the target domain to have perfectly matching domain prototypes. Instead, we assume that the target domains are entirely new and unseen, aiming to anchor their position using existing domain prototypes. To achieve this, we selected checkpoints with diverse styles as our training data to maximize the adaptability of the prompt representations. Consequently, we do not need to continually increase the number of domain prototypes.
>
>
> Q3. If data could have introduced some biases in favor of the proposed method, leading to better results.
> A3. Our basic assumption is that all checkpoints must be with the same structure and fine-tuned based on the same base SD model using different data. Our method addresses the generalization of automatic prompts across different checkpoints. If the target model is with entirely different structure from the training model, such as SD XL or DALLE, the problem itself becomes unsolvable, as we cannot predict the behavior of an entirely unknown model. Generalization is only possible on checkpoints derived from the same model but fine-tuned with different data.

---

> > ### Comment · Area_Chair_RA3c · 2024-08-13
> > **Please respond reviewer redB**
> >
> > Reviewer redB:
> >
> > Please let us know whether the authors have addressed your concerns.
> >
> > Thanks.
> >
> > -AC

---

> > ### Comment · Reviewer_redB · 2024-08-14
> >
> > Thank you to the authors for their responses. They have addressed some of my concerns. However, I still find the performance improvement of the proposed method over the baselines to be marginal, especially considering the significant difference when compared to manual prompting, particularly in terms of Blipscore and ImageReward. Most of all, I am concerned about the fairness of the comparison with other baselines, as the characteristics of effective prompts can be dependent on the choice of diffusion models and the content and style of the targeted images. In the present setting, the data used for training and testing are consistent in that they are all collected from CIVITAI and fine-tuned based on SD1.5. However, the proposed method is the only method optimized for this specific setting, whereas the other baselines are not. This could give a comparative advantage to the proposed method, making it difficult to assert that the evaluation is fair. For these reasons, I maintain my previous rating.

---

> > > ### Author Response · Authors · 2024-08-14
> > >
> > > Dear Reviewer,
> > >
> > > We sincerely appreciate your valuable comments and the recognition of our rebuttal efforts. We would like to further clarify on your mentioned two issues:
> > >
> > > 1. **Regarding the good performance of manual prompting**, this can be attributed to three main factors: iterative selection of prompts (a repeated process of prompt adjustment and image generation), iterative selection of generated images (a repeated process of selecting images over multiple generations), and the possible use of auxiliary tools like ControlNet. In contrast, our method and other baselines do not involve such selection processes or the use of auxiliary tools.
> > >
> > > 2. **Regarding the fairness of the comparison**, this can be explained from three aspects:
> > >    - Firstly, both our method and other baselines are trained on our collected data. For example, "Prompt Perfect" is trained with the data from the checkpoint DreamShaper.
> > >    - Secondly, although all these data are collected from CIVITAI, they are uploaded by different users and generated using private fine-tuned checkpoints. Our experiments also show that data from different checkpoints exhibit significant domain differences and are not from the same distribution. Therefore, the data used for training and testing are not consistent.
> > >    - Thirdly, our goal is to train an adapter on multiple checkpoints to achieve better generalization on unseen checkpoints. However, existing baseline methods cannot accomplish this and fail to produce an adapter with sufficient generalization capability. Therefore, the difference between the two training settings (as shown in Figure 1) also highlights how our method addresses the shortcomings of previous baseline methods, meeting the need to avoid retraining the adapter for new checkpoints.

---

> > > > ### Comment · Reviewer_redB · 2024-08-14
> > > >
> > > > Thank you for the additional information. I will check and take it into account for the final recommendation.

---

### Decision · Program_Chairs · 2024-09-25

**Decision:**

Accept (poster)

**Comment:**

The final ratings are 5, 6, 6. However, it seems there are still some concerns as follow:

- **AP-Adaptor improvement over baselines are marginal**. Reviewer did mention that the performance is far from manual prompting. However, the AC's opinion is that the paper is on improving over automatic prompting so this is not such a big issue to him. Automatic prompt will not beat manual prompting at this time for sure but the research must still go on. So, the AC is not that concerned about achieving "commercial grade" for automatic prompting yet.
- **AP-Adaptor may not be truly out-of-domain capable**. This stems from the concerns that the approach is simply a result of a bunch of diversified checkpoints that allow AP-Adaptor to be multi-domain.
- **The proposed method is heavily dependent on the quality of the prompt generated by the LLM**. The authors tried to address these concerns by proposing two solutions but clearly the solutions are just hypothetical -- the AC don't find this very satisfying in his opinion.

Upon further discussions, reviewers remain a bit borderline on the second point and the third point. On (2), reviewers raised the point of whether the proposed approach is genuinely out-of-domain capable and whether the evaluation setting is entirely fair.

After careful considerations, the AC urged the authors to address the second and third point strongly in the final manuscript and decided to accept the paper as the reviewers finally are slightly more positive than negative about the paper.